# Ablation-resistant carbide $Zr_{0.8}Ti_{0.2}C_{0.74}B_{0.26}$ for oxidizing environments up to 3,000 °C

Yi Zeng[1,2], Dini Wang[1], Xiang Xiong[1], Xun Zhang[2], Philip J. Withers[2], Wei Sun[1], Matthew Smith[2], Mingwen Bai[2] & Ping Xiao[2]

Ultra-high temperature ceramics are desirable for applications in the hypersonic vehicle, rockets, re-entry spacecraft and defence sectors, but few materials can currently satisfy the associated high temperature ablation requirements. Here we design and fabricate a carbide ($Zr_{0.8}Ti_{0.2}C_{0.74}B_{0.26}$) coating by reactive melt infiltration and pack cementation onto a C/C composite. It displays superior ablation resistance at temperatures from 2,000–3,000 °C, compared to existing ultra-high temperature ceramics (for example, a rate of material loss over 12 times better than conventional zirconium carbide at 2,500 °C). The carbide is a substitutional solid solution of Zr–Ti containing carbon vacancies that are randomly occupied by boron atoms. The sealing ability of the ceramic's oxides, slow oxygen diffusion and a dense and gradient distribution of ceramic result in much slower loss of protective oxide layers formed during ablation than other ceramic systems, leading to the superior ablation resistance.

[1] State Key Laboratory of Powder Metallurgy, Central South University, Changsha 410083, China. [2] School of Materials, University of Manchester, Manchester M13 9PL, UK. Correspondence and requests for materials should be addressed to X.X. (email: Xiong228@sina.com) or to P.X. (email: P.xiao@manchester.ac.uk).

Future hypersonic aerospace vehicles offer the potential of a step jump in transit speeds. Currently, one of the biggest challenges is how to protect critical components such as leading edges, combustors and nose tips so that they survive the severe oxidation and extreme scouring of heat fluxes at temperatures in excess of 2,000 °C during flight[1,2]. The diborides of Hf and Zr are considered to be the most promising candidates for such components[3,4], offering the best oxidation resistance up to 1,500 °C among candidate ultra-high temperature ceramics (UHTCs)[5]. In particular, $ZrB_2$ has attracted much attention due to its low density and cost[6–8]. However, there are two critical factors hindering its application: first, a high level of boron (about 66 at. %) leads to severe loss of material under the scouring of hot gas because of the rapid evaporation of boron oxides at temperatures above 1,200 °C (refs 9,10), second, monolithic $ZrB_2$ tends to fail catastrophically due to a combination of low toughness and poor thermal shock resistance[11].

To reduce the evaporation of boron oxides and to improve $ZrB_2$ ablation resistance, much attention has been focused on adding silicides (for example, SiC[12,13], $MoSi_2$ (ref. 14) and so on) and carbides (for example, ZrC[15]) to $ZrB_2$ to form multi-phase ceramics. By contrast relatively little attention has been directed towards the development of a single-phase ceramic comprising multiple elements. In particular, quaternary carbides with low boron contents for ablation resistance have not been reported since UHTCs were first proposed in 1930s[16]. Moreover, although the oxide of ZrC evaporates less at higher temperature due to the absence of boron, it is generally believed that monolithic zirconium carbide (for example, ZrC) has inferior oxidation resistance compared to the diborides (for example, $ZrB_2$) making it a poor option for anti-ablation applications[17–19]. All of the above factors mean that the current number of candidate UHTCs for use in extreme environments is limited and it is worthwhile to explore the potential of new single-phase ceramics in terms of reduced evaporation and better oxidation resistance. In addition, it has been shown that introducing such ceramics into carbon-fibre-reinforced carbon matrix (C/C) composites may be an effective way of improving thermal-shock resistance[20,21].

Here a coating of the quaternary carbide, $Zr_{0.8}$-$Ti_{0.2}$-$C_{0.74}$-$B_{0.26}$, laid down on a C/C composite by reactive melt infiltration (RMI) and pack cementation (PC) is proposed (Methods section and Supplementary Fig.1). The carbide comprises a substitutional solid solution with low boron content. In addition, to improve the thermal-shock resistance, and to decrease the risk of cracking of the carbide coating during ablation, some carbides are allowed to infiltrate into the C/C composite. The experimental results presented here suggest the carbide coating displays better ablation resistance at 2,000–3,000 °C than existing candidate UHTCs such as Zr-based carbide and diborides and other high temperature composites. More broadly, this work provides a platform for building a series of UHTCs based around the group IV/V transition metals.

## Results

**A profile of ablation performance.** Figure 1 compares the ablation resistance of $Zr_{0.8}Ti_{0.2}C_{0.74}B_{0.26}$ coating on C/C composite alongside other common UHTCs and composites. The mass ablation rate (MAR) and linear ablation rate (LAR) characterize the mass loss and dimensional stability of the materials, respectively. In general, a high MAR (that is, rapid loss of mass) and LAR (that is, rapid degradation of surface integrity) indicate poor ablation performance. Hence, the MAR and LAR results for our carbide in Fig. 1 demonstrate a significant improvement in ablation resistance relative to existing UHTCs coatings or composites as well as monolithic $ZrB_2$–SiC ceramics

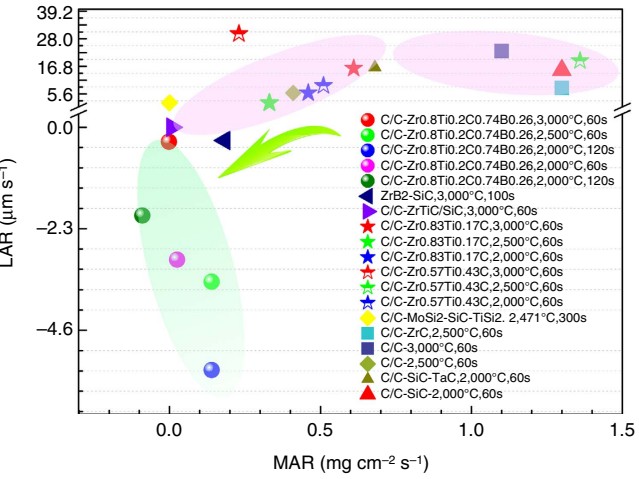

**Figure 1 | Ablation performance of $Zr_{0.8}Ti_{0.2}C_{0.74}B_{0.26}$ composites.** Comparison of the ablation rates (MAR and LAR) for a range of candidate UHTC composites including: ZrC (RMI)[29], SiC (RMI), SiC-TaC (CVI)[42], $Zr_{0.83}Ti_{0.17}C$ (RMI)[29], $Zr_{0.57}Ti_{0.43}C$ (RMI)[29], ZrTiC-SiC (RMI), $MoSi_2$-SiC-$TiSi_2$ (RMI)[43], $ZrB_2$-SiC (SPS) and C/C composites (CVI), as well as the $Zr_{0.8}Ti_{0.2}C_{0.74}B_{0.26}$ studied here (fabrication methods shown in brackets). Above ablation tests were conducted by authors in oxyacetylene machine.

fabricated by spark plasma sintering (SPS). For instance, the LAR of ZrC at 2,500 °C is 8.0 µm s$^{-1}$ and MAR is 1.10 mg cm$^{-2}$ s$^{-1}$, whereas our carbide gains 3.5 µm s$^{-1}$ in thickness and 0.14 mg cm$^{-2}$ s$^{-1}$ in weight. This is because the oxide layer expands and increases the weight countering any material loss from ablation, indicating an almost negligible loss of our carbide. This is over 12 times better than the loss of material for ZrC (assuming both carbides have the same volume expansion of oxides per second). It is noteworthy that the MARs of our carbide from 60–120 s at 2,000–2,500 °C distribute around the zero, indicating a slight weigh loss or weigh gain. These higher negative values of LARs obviously distinguish it from other UHTCs indicated by the green arrows in Fig. 1, indicating that a good quality protective oxide layer of carbide is formed during the ablation experiment (Protective mechanisms section). This layer is strongly adhered to the C/C composite substrate being able to endure the scouring of the high speed hot gas and providing a high level of protection to the substrate (Fig. 3). At 3,000 °C, our carbide still exhibits low LAR and MAR. In addition, it should be noted here that the ablation performance can be mainly attributed to the ceramics, regarding the RMI/chemical vapour infiltration (CVI) process (Methods section and Supplementary Fig.1 and Supplementary Note 1). Generally, a ceramic coating would remain on the composites fabricated by the above methods. Once the coating has been depleted, the carbon matrix would be exposed and be detached quickly by hot gas, causing a very high LAR (see the significance of change for LAR of the composites in Fig. 1). However, the MAR change would be less because of the weight gains of oxides from the ceramics.

Figure 2 shows a photograph of the ablation test and the morphology of the tested sample. Despite the low level of boron present, our carbide displayed the characteristic light green flame during the ablation test, which in contrast to the carbide (that is, $Zr_{0.8}Ti_{0.2}C$ fabricated by RMI), and typical of the ablation flame of $ZrB_2$ (ref. 22), as shown in Fig. 2a. Generally, borides (for example, $ZrB_2$) show better levels of oxidation resistance than their carbides (for example, ZrC)[23]. From the comparison between $Zr_{0.8}Ti_{0.2}C_{0.74}B_{0.26}$ and $Zr_{0.8}Ti_{0.2}C$ shown in Fig. 1, it is inferred that the improvement in ablation resistance can be

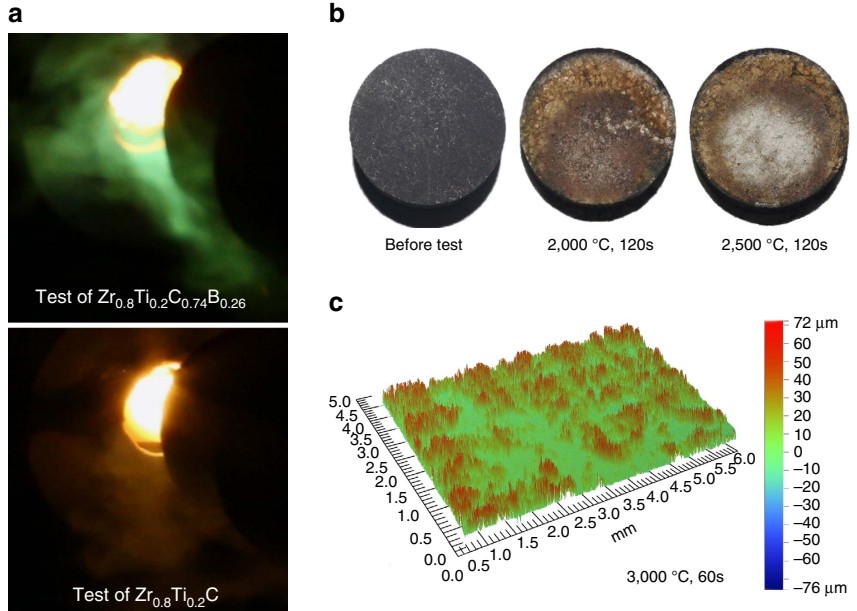

**Figure 2 | Photograph of ablation test and morphology of tested sample.** (**a**) Green and orange flames are seen in tests of $Zr_{0.8}Ti_{0.2}C_{0.74}B_{0.26}$ and $Zr_{0.8}Ti_{0.2}C$, respectively. (**b**) Comparison of surface of the 30 mm diameter samples before and after ablation. Black-gray sample is before test, and middle and right samples experienced 120 s ablation of 2,000 °C and 2,500 °C, respectively. (**c**) Surface profile of central region of sample ablated at 3,000 °C, showing the ablated traces (some convexities with the rises of $<72\,\mu m$) due to evaporations of oxides with low melting points (see Protective mechanisms). But no ablated hollows appeared on surface.

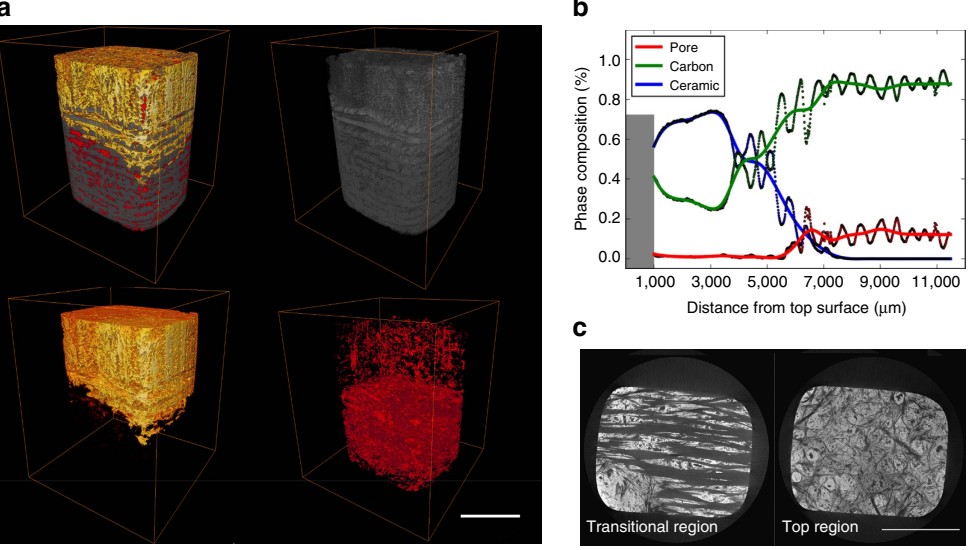

**Figure 3 | Morphology of C/C-$Zr_{0.8}Ti_{0.2}C_{0.74}B_{0.26}$/SiC via X-ray computed-tomography.** (**a**) Overall view of composites showing carbides (yellow), carbon fibres and pyrocarbon (grey) and pores (red). Due to the resolution limit, only pores larger than $1,000\,\mu m^3$ are quantified. (**b**) Distributions of the phases from the top to the bottom of the sample determined from the X-ray computed tomography (CT) virtual slices. The dots are the volume percentage of each phase calculated from each virtual slice. The solid lines are the fitted curves. The zig-zag shape of the dot is due to the alternating plies. The grey area represents the top of the sample which has been excluded from quantification due to strong artefacts. (**c**) Virtual CT cross-sections showing the transitional region comprising the carbon matrix and ceramic (40 vol. %) and a region near the top (ceramic: 76 vol. %). Scale bar, 5 mm.

mainly attributed to the introduction of boron into $Zr_{0.8}Ti_{0.2}C$. In addition, as shown in Fig. 2b,c, the ablated surfaces are relatively smooth and free from any obvious erosion hollows and cracks. The protective oxide layer grows with the increasing temperature remaining essentially intact throughout. Consequently, the $Zr_{0.8}Ti_{0.2}C_{0.74}B_{0.26}$ carbide exhibits a level of ablation resistance and protection not seen in other Zr-based carbides and diborides and common high-temperature composites. The results also

suggest that C/C composite modified by $Zr_{0.8}Ti_{0.2}C_{0.74}B_{0.26}$ (Figs 2 and 3) displays good thermal-shock resistance.

**Microstructure and constituents.** Figure 3 shows the distribution of the $Zr_{0.8}Ti_{0.2}C_{0.74}B_{0.26}$ ceramic, the carbon (carbon fibre and pyrocarbon), and the pores below the surface into the test-piece. The surface region comprises up to 75% ceramic and 25% C. It is

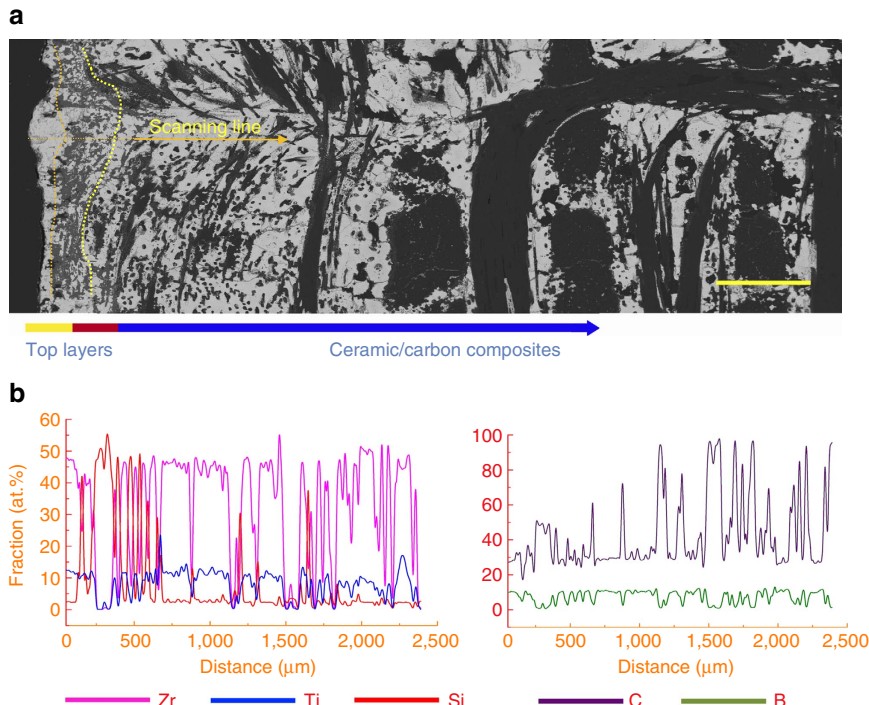

**Figure 4 | Microstructure and element distribution of carbides.** (**a**) SEM image of cross-section of top ceramic layers and ceramic/carbon composites. From left to right, the external layer is the carbide with the thickness of 100–200 µm, the second layer composed of the carbide and SiC with the thickness of 200–300 µm. Beneath that there are the ceramic and carbon composites as indicated by the arrows. (**b**) Electron probe microanalysis reveals element distributions. Lines of pink, blue, red, purple and olive lines represent Zr, Ti, Si, C and B, respectively. In the ceramic areas (white areas in **a**), the ratios of Zr, Ti, C and B are about 40, 10, 37 and 13 at. %, respectively. Grey areas beneath the external layer are SiC according to results of the scan. Scale bar, 500 µm.

sufficiently dense that the top surface (see cross-section in Fig. 4) acts as a barrier to resist oxidation and scouring from the hot gas during the ablation test. The porosity (pore size $>10\,\mu m$) is $<3\%$ and the volume of ceramic is higher than 58 % near the free surface shown in Fig. 3b,c (top region). Deeper into the sample (7,000 µm), it comprises carbon and pores (shown in Fig. 3a,b). In these areas (carbon-based part), the volumes of carbon and pores reach 90 and 10%, respectively, with no ceramic present. However, in the transitional region shown in Fig. 3c, the ceramic and carbon show a smooth gradient distribution with no sharp interface between the ceramic and the C/C composite. Generally, the C/C composite and Zr-based ceramics have significantly different coefficient of thermal expansion (CTE) (for example, $CTE_{C/C} = 0.38–2.18 \times 10^{-6}\,C^{-1}$ (ref. 24), $CTE_{ZrC} = 6.70 \times 10^{-6}\,C^{-1}$ (ref. 25) and $CTE_{ZrB2} = 6.66–6.93 \times 10^{-6}\,C^{-1}$ (ref. 26)). Thus the gradient distribution employed in this work, together with the fibre reinforcement and weakened pyrocarbon interfaces[27,28] might alleviate the mismatch in CTE, leading to an improvement of thermal shock resistance and the density of whole sample because of the presence of the carbon-based part.

Figure 4 shows a cross-section through the top layer of carbide. The ablation resistant layer composed of $Zr_{0.8}Ti_{0.2}C_{0.74}B_{0.26}$ is about 100–200 µm in thickness. Beneath the carbide layer, SiC is identified which has formed from the reaction between the original carbon and the Si/SiO during PC process (Methods section and Supplementary Note 2), as shown in Fig. 4a. The formula of $Zr_{0.8}Ti_{0.2}C_{0.74}B_{0.26}$ carbide was obtained, according to elemental analysis conducted by electron probe microanalysis (EPMA) as shown in Fig. 4b. The ratio of Zr and Ti is dictated by the raw powders (80 at. %:20 at. %) which was optimized for ablation resistance at temperatures over 2,000 °C from our previous investigation on the doping effect of Ti in Zr[29]. The ratio of B to C was 0.74:0.26 according to the EPMA results which is discussed in detail in the following section.

Actually, the formation of our carbide forms in three stages. First, the Zr-Ti melt infiltrates the porous C/C composite at high temperature and reacts with the pre-deposited pyrocarbon[30] to form $Zr_{0.8}Ti_{0.2}C_{(1-x)}$ $(0<x<1)$ (Methods section and Supplementary Fig. 2). Subsequently, the $Zr_{0.8}Ti_{0.2}C_{0.74}B_{0.26}$ forms through the boration of $Zr_{0.8}Ti_{0.2}C_{(1-x)}$ via solid diffusion of boron atoms (see following reactions) during PC process. $Zr_{0.8}Ti_{0.2}C_{(1-x)}$ is a substitutional solid solution which has a FCC structure along with carbon vacancies (Fig. 5b,e). In this process, the boron atoms fill the original vacancies replacing the carbon by diffusion, which has not changed the stacking type of atoms of Zr-Ti carbide. This interpretation is substantiated by ZrC diffraction peaks, instead of $ZrB_2$, obtained from the top surface and cross-section of the composites shown in Fig. 5a, which has been also confirmed by the high resolution TEM images and their diffraction patterns of FCC shown in Fig. 5e,f. However, the replacement of interstitial atoms causes a variation in the crystal lattice constant (a) shown in Fig. 5b,e and f. For instance, the absence of carbon atom and substitution of Zr by Ti in $Zr_{0.8}Ti_{0.2}C_{(1-x)}$ result in a smaller a, as shown in Fig. 5b, compared with pure ZrC. Nevertheless, the vacancies were occupied by the boron atoms leading to a slight increase of a.

Consequently, the structural changes in Zr-Ti-C-B carbide can be described using the schematic representation shown in Fig. 5c,d. The defect channels in the FCC crystals of $Zr_{0.8}Ti_{0.2}C_{(1-x)}$ have been built by disordered carbon vacancies (possibly having short-range-order[31], Fig. 5c). The small boron atoms diffuse relatively quickly through the crystal boundaries, interstices and the defect channels to preferentially occupy the vacancies originally occupied by carbon atoms (Fig. 5d), due to the lower formation energy of a carbon vacancy[31]. This suggests that the large extent of the boron distribution in the carbide shown in Fig. 4 is possibly attributed to the pre-existing crystal defects that greatly promoted the diffusion of boron atoms. It is

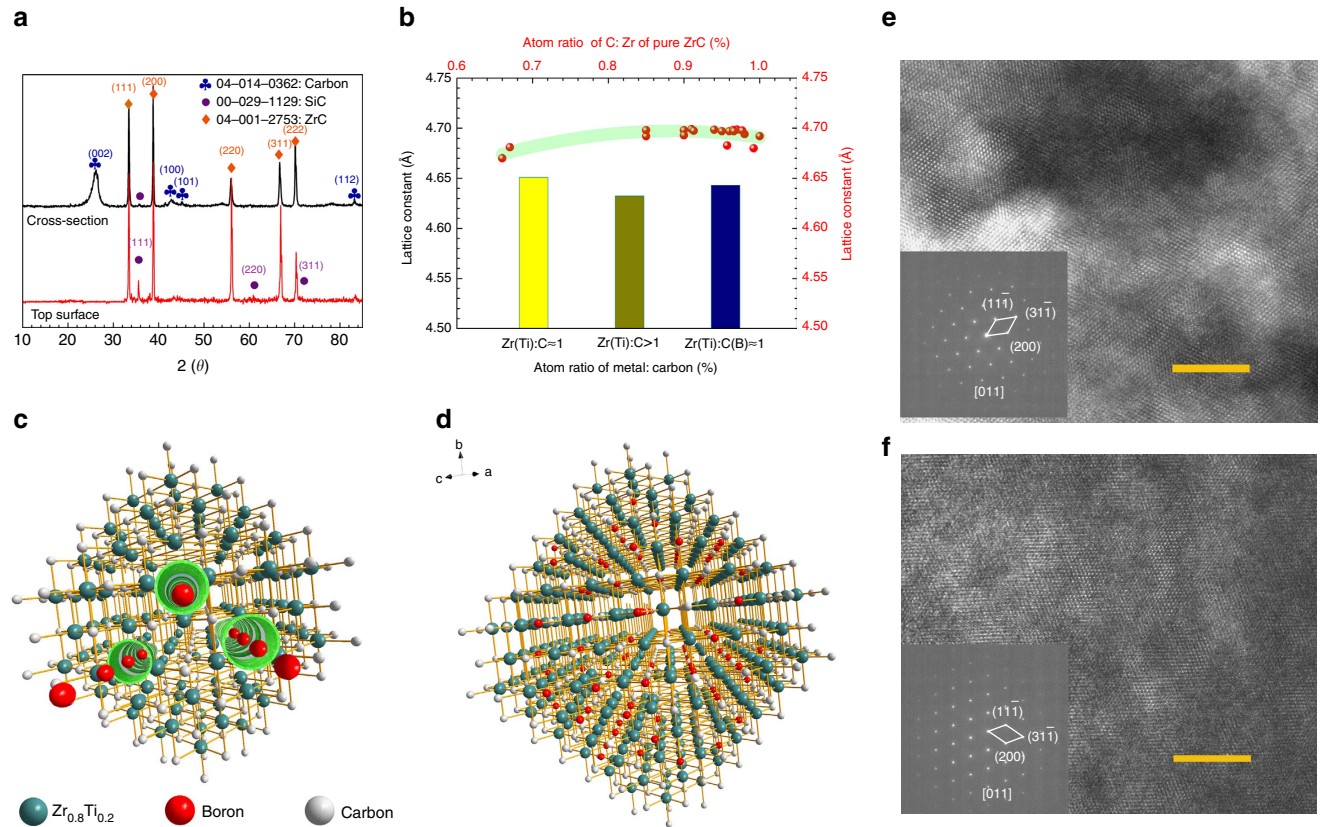

**Figure 5 | Phases and structural changes in the Zr-Ti-C-B carbide during the PC process.** (**a**) XRD results of top surface and cross-section of composites. Black curve is from cross-section which is mainly composed of ZrC, carbon and a small amount of SiC. Red curve is from top surface which is mainly composed of ZrC and SiC. However, TiC and diborides have not been observed due to the substitution of 20 at. % Zr by Ti and the vacancy filling of boron atoms. (**b**) Variation of lattice constant in carbide crystal. The measurement of lattice constant is from the Rietveld refinement of XRD results (Methods section). Red dots representing lattices of pure ZrC are from ref. 44. Columns are the lattice constants of carbide in this work. (**c**) Schematic representation of distribution of boron atoms in $Zr_{0.8}Ti_{0.2}C_{(1-x)}$ through defect channels. (**d**) Schematic representation of boron atoms filling in the carbon vacancies. (**e**) High resolution transmission electron microscope (TEM) image of $Zr_{0.8}Ti_{0.2}C_{(1-x)}$. The inset in **e** is the electron diffraction pattern of $Zr_{0.8}Ti_{0.2}C_{(1-x)}$. (**f**) High resolution TEM image of $Zr_{0.8}Ti_{0.2}C_{0.74}B_{0.26}$ from the focused ion beam (FIB) samples. (Methods section and Supplementary Fig. 3). The inset in **f** is the electron diffraction pattern of $Zr_{0.8}Ti_{0.2}C_{0.74}B_{0.26}$. Scale bar, 5 nm.

also inferred that, due to the pre-synthesized stable FCC structure during the RMI, the $Zr_{0.8}Ti_{0.2}C_{(1-x)}$ has been inhibited to transform into hexagonal structure of crystals such as $ZrB_2$ and $TiB_2$ during the subsequent reactions.

Consequently, according to the variations of structure and phases in $Zr_{0.8}Ti_{0.2}C_{0.74}B_{0.26}$, as well as the raw materials, we conclude that the following boration reactions occurred during the PC process.

$$14Zr_{0.8(1-x)}Ti_{0.2(1-x)}C_{(1-x)} \cdot x V'_C Zr_{0.8}Ti_{0.2(s)} + 3xB_4C_{(s)} + xB_2O_{3(l.g)} =$$
$$14Zr_{0.8(1-x)}Ti_{0.2(1-x)}C_{(1-x)} \cdot x(Zr_{0.8}Ti_{0.2} \cdot B)_{(s)} + 3xCO_{(g)} =$$
$$14Zr_{0.8}Ti_{0.2}C_{(1-x)}B_{x(s)} + 3xCO_{(g)}.$$

$$\text{(1)}$$

$$7Ti(Zr)_{(s.l)} + 3B_4C_{(s)} + B_2O_{3(l.g)}$$
$$= 7Zr(Ti)B_{2(s)} + 3CO_{(g)}.$$

$$\text{(2)}$$

$$7C_{(s)} + 2B_2O_{3(l.g)} = B_4C_{(s)} + 6CO_{(g)}.$$

$$\text{(3)}$$

where $V'_C$ and $x \cdot Zr_{0.8}Ti_{0.2}$ are the vacancy of carbon atoms and extra metal atoms in non-stoichiometric carbide, respectively. In reaction (1), $Zr_{0.8(1-x)}Ti_{0.2(1-x)}C_{(1-x)} \cdot x V'_C \cdot xZr_{0.8}Ti_{0.2}$, is equivalent to $Zr_{0.8}Ti_{0.2}C_{(1-x)}$ formed from the RMI process. $B_4C$ and $B_2O_3$ are the raw materials and $Zr_{0.8}Ti_{0.2}C_{(1-x)}B_x$ is the final

product. In this work, the average content of boron is about 13 at. % in the carbide and thereby $x$ is 0.26 here. Hence, reaction (1) governs the main reaction of boron atoms. $V'_C$ finally disappeared because of the occupation of boron atoms. Moreover, the residual metal such as $Zr^{32}$ and Ti in the composite would further react with the raw materials to form diborides (see the diborides observed by EPMA in Supplementary Fig. 4), as shown in reaction (2). Actually, the doped $B_2O_3$ may react with carbon fibres and pyrocarbon, as shown in reaction (3), which is another source of the $B_4C$ in reaction (1). Reaction thermodynamics are referenced in Supplementary Fig. 5.

**Protective mechanisms.** Figure 6 shows the surface and cross-sectional microstructure at the centre of the ablated surface and the phases across the whole ablated surface. At 2,000 °C, $Zr_{0.8}Ti_{0.2}C_{0.74}B_{0.26}$ and SiC are oxidized and converted into $Zr_{0.80}T_{0.20}O_2$, $B_2O_3$ and $SiO_2$, respectively, as shown in Fig. 6a,e. $Zr_{0.80}Ti_{0.20}O_2$ partially melts and forms a relatively dense layer in the central ablated surface. However, the evaporants with low melting point, such as $SiO_2$ and $B_2O_3$, escape from the oxide layer leaving evacuation channels: the holes, as shown in Fig. 6a. At a higher ablation temperature (2,500 °C), the $Zr_{0.80}Ti_{0.20}O_2$ crystals connected by the melt become larger and the holes evidently shrink as shown in Figure 6b. Its cross section shows

the porous morphology under the dense surface, as shown in Fig. 6f. Obviously, three different layers can be observed on the cross-section: porous external layer, intermediate layer and dense inner layer, possibly due to the evaporation of oxides and

thermal gradient perpendicular to oxide layer. For instance, higher temperature would result in more severe evaporation at the position closed to the external surface. However, the size of pores in the layers decreases significantly from the external layer

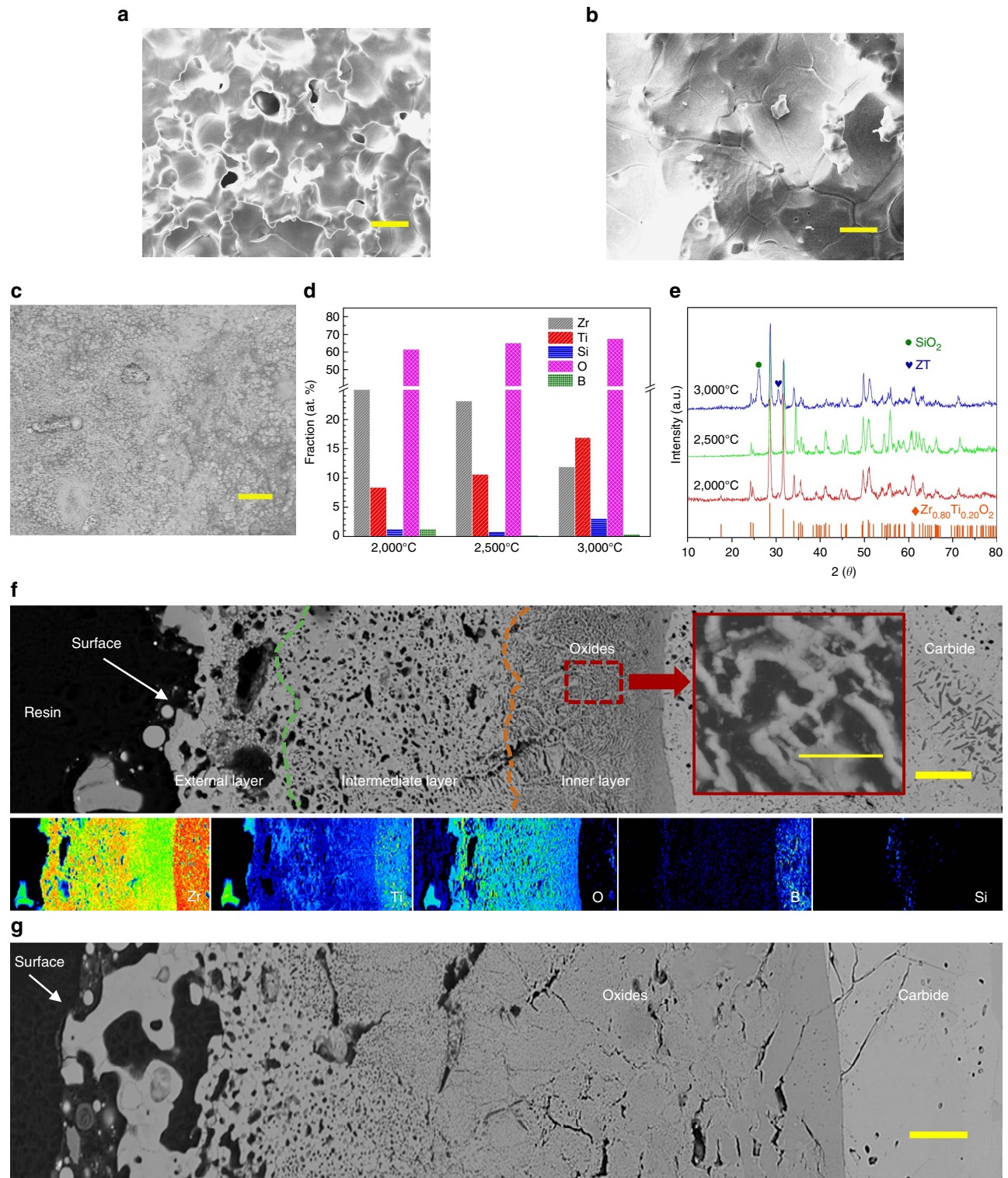

**Figure 6 | Microstructure and phases of ablated surface and cross-section.** (**a**) Central surface ablated at 2,000 °C. (**b**) Central surface ablated at 2,500 °C. (**c**) Central surface ablated at 3,000 °C. (**d**) Concentration of elements in central ablated surface. (**e**) XRD results of whole ablated surfaces. ZT is zirconium titanate having a-PbO$_2$ structure[45]. (**f**) Cross-section morphology of central ablated point at 2,500 °C (back scattered electron images), and the associated distribution of elements. The inset in (**f**) shows a higher magnification of inner layer. (**g**) Cross-section morphology of central ablated point at 3,000 °C. Scale bar in **a,b,c**, 20 μm. Scale bar in **f,g**, 40 μm (inset, 2 μm).

to the inner layer. Especially, the inner layer composed of the grain skeletons and amorphous phases displays a very dense morphology (see inset in Fig. 6f). It is inferred that the dense surface and inner layer act as a barrier to resist oxidation and result in reduced loss of oxide and the best ablation resistance shown in Fig. 1. At 3,000 °C, an almost fully dense oxide layer, mainly composed of $Zr_{0.80}Ti_{0.20}O_2$, zirconium titanate and $SiO_2$, is formed as shown in Fig. 6c,e. In addition, the XRD results suggest the presence of more amorphous phases, as the ablation temperature increases from 2,000 to 3,000 °C, due to the quenching of more liquid-solid phases of oxide layer at the end of ablation test (some oxides under the surface may remain solid at 3,000 °C within the limited ablation time, due to the thermal barrier provided by the Zr-O-Ti ceramic system[33]). Meanwhile, the sealing of the oxides can be attributed to these melts having a relative lower viscosity. However, such melts seal the defects (holes and cracks arising from the ablation) and protect the carbon matrix well, causing greater loss of the oxides by the scouring of hot high-speed gas as well, which is confirmed by the relatively low weight gains shown in Fig. 1 and the corrosion pores beneath the dense surface shown by the cross-section morphology of the sample after 3,000 °C ablation in Fig. 6g. However, a relatively dense inner layer located at the interface between oxide layer and carbide acted a barrier to the diffusion of oxygen, though some pores occurred in a thicker oxide layer formed with the temperature increasing from 2,500 to 3,000 °C. In addition, micro-cracks occurred on the cross-sections due to the thermal shock during the ablation test. The relatively integrated interfaces between oxide layers and carbides experiencing 2,500 and 3,000 °C ablation tests, showing a compact morphology without separation, indicate a good adhesion of oxide layers to substrates. Notably, it is believed that the dense $Zr_{0.80}T_{0.20}O_2$ layer has effectively retained the boron and silicon and extended their consumption time, according to their residual contents shown in Fig. 6d,f.

## Discussion

The carbide presented here exhibits superior ablation response compared to current UHTCs. This can be attributed to the following facts. First, a relatively dense oxide layer with a suitable viscosity plays a key role in resisting the ablation of extremely hot gas. Generally, loose scale with a very high viscosity (for example, no melt), or a liquid layer with a very low viscosity (for example, freely flowing), provides poor protection because of the tendency for detaching or splashing of oxides[29], respectively. In this work, $Zr_{0.80}Ti_{0.20}O_2$ displays a durable and robust structure comprising grain skeletons and liquid phases as shown in Fig. 6, which effectively decreases the loss of oxides, and the dense oxide layer displays good sealing protection even at 3,000 °C. In particular, the doping of Ti as the second main phase led to the formation of the $Zr_{0.80}Ti_{0.20}O_2$ during the ablation test and decreased the viscosity of pure $ZrO_2$ melt due to the lower melting point of $TiO_2$ (1,843 °C) than that of $ZrO_2$ (2,715 °C), which conferred a self-healing ability on the oxide layer, instead of a porous layer. Meanwhile this evidently decreased the vaporization and loss rate of oxides, compared with the conventional $ZrB_2$–SiC or ZrC–SiC ceramic systems. For instance, the vaporization rate (VR) of $TiO_2$ is $0.23\,mm\,s^{-1}$ at 2,227 °C, whereas the VR of $SiO_2$ ($207\,mm\,s^{-1}$) is around 900 times higher than that of $TiO_2$ at the same temperature[34]. This suggests that oxidation of $Zr_{0.8}Ti_{0.2}C_{0.74}B_{0.26}$ will result in much less loss of oxide because of vaporization and good adhesion to the substrate shown in Fig. 6g even at 3,000 °C, compared to the $ZrO_2$-$SiO_2$ system from the oxidation of $ZrB_2$-SiC and ZrC-SiC. Second, the good ablation response is due to the low oxygen permeability

(OP) of the oxide layer. In this work, the $Zr_{0.80}Ti_{0.20}O_2$ layer on the quaternary carbide has effectively caught the $B_2O_3$ and $SiO_2$ which has a very low OP ($OP_{B2O3} = 8.6 \times 10^{-12}\,g\,cm^{-1}\,s^{-1}$ at 1,000 °C, $OP_{SiO2} = 3.2 \times 10^{-15}\,g\,cm^{-1}\,s^{-1}$ at 1,000 °C)[35] and, to a certain extent, can prevent the fast diffusion of oxygen atoms into $Zr_{0.8}Ti_{0.2}C_{0.74}B_{0.26}$ at different ablation temperatures. Moreover, it is believed that the intrinsic oxygen diffusion coefficient (ODC) of the $Zr_{0.80}T_{0.20}O_2$ layer is lower than that of the pure $ZrO_2$ due to the lower ODC of $TiO_2$ ($1.12 \times 10^{-13}\,m^2\,s^{-1}$, at 1,800 °C) compared with $ZrO_2$ ($ODC_{ZrO2} = 1.16 \times 10^{-12}\,m^2\,s^{-1}$, at 1,800 °C)[36]. The lower oxygen permeability of $Zr_{0.80}Ti_{0.20}O_2$ leads to less formation and loss of oxides, and further improves the ablation resistance of carbide, though the melting point of oxide ($Zr_{0.80}Ti_{0.20}O_2$) is lower than 3,000 °C. Consequently, it is believed that our carbide displaying a better ablation resistance than the conventional ceramics can increase the survival time of the components in extremely oxidizing environments up to 3,000 °C. In addition, the lower boron content in $Zr_{0.8}Ti_{0.2}C_{0.74}B_{0.26}$ compared with that in $ZrB_2$ leads to a lower mass loss and fewer defects such as pores and cracks originating from the evacuation channels of $B_2O_3$ at higher temperature, which is significantly beneficial to the ablation response. Third, it also benefits from the gradual transition from ceramic to carbon composite. A dense and high volume of ceramic is more conducive to forming a dense oxide layer and ensured the minimum damage to the underlying carbon matrix from the extreme hot and oxidizing gas.

In conclusion, we have designed a carbide assembled by solid solution and atomic diffusion during PC and RMI. Importantly, the ceramic layer displays better ablation resistance (eg, a rate of material loss over 12 times better than conventional ZrC at 2,500 °C) under oxidizing environments up to 3,000 °C relative to existing common UHTCs and high temperature composites. More broadly, this work provides a platform for building a series of such UHTCs (eg, A (M) $C_xB_y$), where A and M are the main atoms (transition metals, IV) and the substitution atoms (transition metals, IV/V), respectively. For instance, $Hf_x$ $(Zr_{y1}/Ti_{y2}/Ta_{y3})C_{0.8}B_{0.2}$, $Zr_x (Ti_{y1}/Ta_{y2})C_{0.8}B_{0.2}$ and $TiC_{0.8}B_{0.2}$ and so on, can be built, according to similar methods. To increase ablation resistance across a wide range of temperatures, a carbide's oxides layer that owns low VR, OP, good self-healing ability and adhesion strength to the substrate can be achieved through the doping contents of different substitution atoms ($y_i$) of ceramic with a variety of melting points. Consequently, we develop an effective means of fabricating ablation resistant UHTCs. Moreover, these ceramics can be fabricated into powders, bulk materials and layers to extend their application. For instance, in addition to the potential use in hypersonic vehicles, it is expected that they can be used as the nozzle throats and diffusers for reusable rockets, which requires a very low ablation rate to extend the lifetime for recyclability at low cost. Other potential uses may include the hot section components in re-entry spacecraft, defence army, gas turbines and nuclear areas and so on.

## Methods

**Materials and preparation.** Supplementary Fig. 1 shows a schematic of the sample preparation process. T700 Polyacrylonitrile-based carbon fibres were employed as the reinforcement and fabricated into a needled integrated preform (NIP). The NIP was fabricated by a three dimensional needling technique, built up by repeatedly overlapping layers of 0° non-woven C-fibre cloth (A in Supplementary Fig. 1a), a chopped fibre web (B in Supplementary Fig. 1a), and 90° non-woven fibre cloth with needle-punching step by step. The bulk preforms were densified to a porous C/C composite of 1.0–1.3 g cm$^{-3}$ density using pyrocarbon deposited by CVI using $CH_4$ and $H_2$ gases at 900–1,000 °C. The open porosity of the composites with NIP ranged from 39.8 % to 28.8 %. For instance, a sample having a density of 1.16 g cm$^{-3}$ and an open porosity of 34.3 %

possessed a modal pore diameter of 42.0 μm shown in Supplementary Fig. 6. Because non-stoichiometric carbide tends to form when carbon reacts with Zr–Ti melt (Supplementary Fig. 2), a $Zr_{0.8}Ti_{0.2}C_{(1-x)}$ carbide was introduced into the C/C composite by RMI in argon at 1,800–2,000 °C for 0.5–2 h. In this step, an optimized ratio of powders mixed as 80 at. % Zr–20 at. % Ti was used[29]. The viscosity of 80 at. % Zr–20 at. % Ti at the temperature of 1,800–2,200 °C ranged from $4.55 \times 10^{-3}$ to $3.06 \times 10^{-3}$ Pa s as shown in Supplementary Fig. 7. The molten metal was drawn along the carbon fibres by capillary forces, reacting with the previously deposited pyrocarbon to form a carbon/carbide composite. The infiltrated depth and the volume of carbide are dependent on the mass of the melt and the volume and size distribution of the open pores in the C/C composite. Finally, a carbide layer was formed on carbide-based part by PC process (Supplementary Fig. 1c). During the PC process, $B_4C$, $B_2O_3$, SiC, Si, carbon and some catalysts such as $Al_2O_3$ were the raw powders. The carbide-based part was packed by powders and placed in a graphite crucible. At 1,600–1,800 °C, these powders react with $Zr_{0.8}Ti_{0.2}C_{(1-x)}$ and carbon fibres and pyrocarbon to form the $Zr_{0.8}Ti_{0.2}C_{0.74}B_{0.26}$ ceramic and SiC, respectively. In addition, to achieve relatively low porosity, the carbon-based part of sample was further densified using CVI. For comparison of ablation performance, C/C-$Zr_{0.8}Ti_{0.2}C$, ZrTiC-SiC and C/C-SiC composites were fabricated by RMI. In addition NIP processed C/C composite having a density of 1.7 g cm$^{-3}$ was densified by CVI. $ZrB_2$-SiC ceramics were prepared by SPS. More details about the fabrication were referenced Supplementary Note 1.

**Ablation testing.** The ablation behaviours of the samples were evaluated using an oxyacetylene flame (Supplementary Movie 2). During the test, the specimen, having a size of ⌀ 30 × 15 mm, was exposed to the flame. The flow rates and pressure of oxygen were respectively 1.96 l s$^{-1}$ and 0.400 MPa, and those of acetylene were 0.696 l s$^{-1}$ and 0.095 MPa, respectively. The normal combustion ratio of oxygen and acetylene is 1.5, according to $2C_2H_2 + 3O_2 = 4CO + 2H_2O$, and in this work the extra oxygen ensured a sufficient combustion of acetylene and established an extreme oxidizing scenario. During the test, an optical pyrometer indicated that the highest temperature of the central ablated surface reached about 3,000, 2,500 and 2,000 °C at the distances of 10, 20, and 30 mm between the torch nozzle and sample surfaces, respectively. The heat fluxes measured by a water-cooled heat flux sensor at 3,000, 2,500 and 2,000 °C are 5.62, 3.86, 2.57 MW m$^{-2}$, respectively. The inner diameter of the oxyacetylene gun tip was 2 mm. The linear ablation rate (LAR) and mass ablation rate (MAR) were calculated according to $\bar{l} = \Delta l/(\Delta t \cdot S)$ and $\dot{m} = \Delta m/(\Delta t \cdot S)$ respectively, where $\bar{l}$ refers to LAR and $\dot{m}$ to MAR; $\Delta l$ and $\Delta m$ are the decrease in length and the mass loss of specimen, respectively, and $\Delta t$ is the ablating time. $S$ is the ablation area. Ablation rate is averaged over three specimens. The tests lasted 60 and 120 s.

**Characterization methods.** 3D X-ray computed tomography (CT) was conducted using a Zeiss Xradia Versa 520 X-ray microscope at the Henry Moseley X-ray Imaging Facility (HMXIF, Manchester, UK). The accelerating voltage and current of the X-ray tube were set as 140 kV and 72 mA, respectively. Each scan comprised 1,601 radiographs taken incrementally over a rotation angle of 360̂. A 3D volume rendering of the sample was created from the virtual slices in AVIZO software. The first 100 slices and last 100 slices were cropped due to low image quality presumably as a result of the cone-beam geometry. Pores, carbon and ceramics were segmented out using the top hat method[37] (Supplementary Movie 1). XRD experiments were carried out on a Panalytical MPD system using Cu radiation. The voltage and current were 45 kV and 40 mA, respectively. For the analysis of phases, the $2\theta$ scan range was 10–90°, scanning resolution was 0.05° per step. For the measurement of crystals structure, the samples were scanned between 10–90° $2\theta$, at a step-width of 0.02° and scan speed of 0.5°·per min. Rietveld refinement was carried out with the program Maud using a pseudo-Voigt profile function[38,39], and Rw was less than 9.6%.

A super probe electron probe microanalysis system (EPMA, JEOL, Jxa8230) was used to detect the content and distribution of elements. High resolution transmission electron microscope (HRTEM) images and selected area electron diffraction (SAED) patterns were obtained with an FEI Talos F200A microscope equipped with an X-FEG electron source. TEM samples were prepared by focused ion beam (FIB, FEI Quanta 3D) using the in-situ lift-out technique on cross-sections of the samples[40]. The morphology of samples was studied by scanning electron microscopy (SEM, FEI, NOVA Nano230). Bulk density was measured according to the Archimedes method[41]. The 3D surface profile was conducted using the 3D Optical Microscopy (Bruker Contour Elite 3D Optical Microscope). The open porosity was measured using the boiling water method according to the ASTM Standard C20-00, and the pore size distribution of NIP samples was investigated using mercury porosimetry (Quantachrome, Pore Master 60), according to ISO 15901-1.

**Data availability.** The authors declare that the data supporting the findings of this study are available from the corresponding authors on reasonable request.

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

## Acknowledgements

We thank National Basic Research Program of China (No.2011CB605805), China Postdoctoral Exchange Fellowship Program (20140012) and National Natural Science Foundation of China (51602349) for funding of this work. The authors would also like to thank the Henry Moseley X-ray Imaging Facility (HMXIF, Manchester, UK) for x-ray tomography measurement funded by the EPSRC through grants EP/M010619, EP/K004530, EP/F007906, EP/F028431. We are indebted to Han Liu in University of Manchester for the assistance with calculation of the thermodynamics.

## Author contributions

X.X., P.X. and Y.Z. proposed and designed the project. Y.Z. and D.W. developed the optimal composition of ceramic and fabricated the ceramic composites and the analyses including SEM, TEM, XRD and EPMA. M.B. carried out the FIB and M.S. conducted the TEM operation and analyses. W.S. and D.W. conducted the ablation test and measurement of property. X.Z. and P.W. carried out the 3D X-ray tomography and the analyses. Y.Z. wrote the paper with input from all authors, and P.W., X.X. and P.X. refined the paper. All authors contributed to the interpretation of the results.

## Additional information

**Competing interests:** The authors declare no competing financial interests.

