## [Peer Review File · Nature Communications]

Reviewers' Comments:

Reviewer #1 (Remarks to the Author):

This manuscript titled “Highly ablation resistant $Zr_{0.8}Ti_{0.2}C_{0.74}B_{0.26}$ carbide for extremely oxidizing environments up to $3000^{\circ}C$ ” by Dr. Xing and co-authors report a new carbide ($Zr_{0.8}Ti_{0.2}C_{0.74}B_{0.26}$) coating by reactive melt infiltration and pack cementation onto a C/C composite. The results showed that the coating after oxidation has good sealing ability due to its low oxygen diffusion and a dense and gradient distribution of ceramic. Although the materials design and results are interesting, the inherent shortcomings should be overcome before its publication.

- (1) During the RMI process, the solution of molten Zr–Ti was infiltrated into the porous C/C composite. However, the author should supply the information of melt-viscosity of Zr–Ti, because the infiltration height is associated to the melt viscosity. Also how the pore size and porosity for the NIP?
- (2) It is suspectable that the $Zr_{0.8}Ti_{0.2}C_{0.74}B_{0.26}$ coating can resist high-temperature to $3000^{\circ}C$. Because ZrO_2 or TiO_2 melting point is below $3000^{\circ}C$. In UHTC system, for ZrB_2 -SiC, can hardly resist to $2800^{\circ}C$ in the oxyacetylene flame environment.
- (3) In the section of ablation mechanism, the authors had better supply cross-section morphology after ablation. It can clearly see the change and variation of oxide layer after ablation.
- (4) With the increase in ablation temperature, more amorphous phases occurred according to the variation of the shape and intensity of XRD peaks and SEM. Why these phenomenon (amorphous phases) happen?

Reviewer #2 (Remarks to the Author):

The claims of the paper are the development of single phase quaternary $ZrTiCB$ compounds for ablation resistance, compounds that haven't been investigated in 80+ years. The ablation data (up to $3000^{\circ}C$) shows a marked improvement in ablation resistance over ZrB_2 or ZrC .

Yes, for those of us in the hypersonic community it will surely be of interest as a promising coating for protection of the underlying structure.

The work is convincing in terms of the data presented. The discussion of the relevance of the work, as well as the interpretation, is overall solid.

I believe this paper will spark interest in investigating other quaternary compounds for hypersonic applications as it represents a “new” promising material class.

I believe the work of the authors is well described in this overall excellent paper. Not merely the presentation of some interesting results, the authors perform a relatively in depth analysis of the new compound. For example, see Figure 6c and d, where atomic placement is shown.

Some comments:

Page 3, line 75 – there is no matching parenthesis.

Page 4 – what is the heat flux of the ablation flame? This is a standard way to compare data.

Also, what opening was used on the gas tip? 3 mm dia? 5 mm dia? Why did the tests last “approximately” 60 secs and 120 secs as opposed to exactly 60 secs or 120 secs? What was the error in time measurement?

Responses to Reviewers:

Reviewers' comments:

Reviewer #1 (Remarks to the Author):

This manuscript titled "Highly ablation resistant Zr_{0.8}Ti_{0.2}C_{0.74}B_{0.26} carbide for extremely oxidizing environments up to 3000°C" by Dr. Xiong and co-authors report a new carbide (Zr_{0.8}Ti_{0.2}C_{0.74}B_{0.26}) coating by reactive melt infiltration and pack cementation onto a C/C composite. The results showed that the coating after oxidation has good sealing ability due to its low oxygen diffusion and a dense and gradient distribution of ceramic. Although the materials design and results are interesting, the inherent shortcomings should be overcome before its publication.

(1) During the RMI process, the solution of molten Zr–Ti was infiltrated into the porous C/C composite. However, the author should supply the information of melt-viscosity of Zr-Ti, because the infiltration height is associated to the melt viscosity. Also how the pore size and porosity for the NIP?

We are happy to clarify in the text. Actually, the viscosities of pure Zr and Ti at 2073-2473K have been measured by Takehiko Ishikawa et al. using an electrostatic levitator. The viscosity of Zr-Ti binary alloy melt was calculated according to Moelwyn-Hughes and Iida methods referenced to the book (Roderick I. L. and Takamichi, IIDA. The physical properties of liquid metals, Oxford University Press, 1988.). Hence, in the Methods section of the revised manuscript, we have added the following content:

"During the RMI process, the solution of molten Zr–Ti was infiltrated into the porous C/C composite. In this step, an optimized ratio of powders mixed as 80 at. % Zr –20 at. %Ti was used²². The viscosity of 80 at. % Zr –20 at. %Ti at the temperature of 1,800-2,200 °C ranged from 4.55×10^{-3} to 3.06×10^{-3} Pa·s shown in Supplementary Fig. 7."

Supplementary Figure 7. Viscosity of Zr, Ti and 80 at. % Zr –20 at. % Ti melts as function of temperature. The viscosities of Zr and Ti were measured by Ishikawa et al², and the viscosity of binary alloy melt of Zr-Ti was calculated according to Moelwyn-Hughes and Iida methods³. More details about the calculation can be found in ref 4.

Supplementary references

- Ishikawa, T., Paradis P.F., Okada J.T., & Watanabe Y. Viscosity measurements of molten refractory metals using an electrostatic levitator. *Meas. Sci. Technol.* 23, 1-9 (2012).
- Roderick I. L. & Takamichi IIDA. *The physical properties of liquid metals*, Oxford University Press, 1988.
- Zeng Y. et al. Preparation and microstructure of carbon/carbon composites modified with Zr-Ti-C fabricated by reacted melt infiltration. *The 8th Pacific Rim international congress on advanced materials and processing, Hawaii, USA, 447-456(2013)*.

Additionally, the porosity and the pore size distribution of NIP samples have been measured according to the ASTM Standard C20-00 and ISO 15901-1, respectively. Their results are shown in Methods and Supplementary Information of the revised manuscript. In the Methods section, we added the following content:

“The bulk preforms were densified to a porous C/C composite of 1.0-1.3 g·cm⁻³ density using pyrocarbon deposited by CVI using CH₄ and H₂ gases at 900–1,000 °C. The open porosity of the composites with NIP ranged from 39.8 % to 28.8 %. For instance, a sample having a density of 1.16 g·cm⁻³ and an open porosity of 34.3 % possessed a mode pore diameter of 42.0 μm shown in Supplementary Fig. 6.”

In the Characterization section, we have added following content:

“The open porosity was measured using the boiling water method according to the ASTM Standard C20-00, and the pore size distribution of NIP samples was investigated using mercury porosimetry (Quantachrome, Pore Master 60), according to ISO 15901-1.”

Supplementary Figure 6. Size distribution of open pores in composite with NIP having a density of $1.16 \text{ g}\cdot\text{cm}^{-3}$.

(2) It is suspectable that the $\text{Zr}_{0.8}\text{Ti}_{0.2}\text{C}_{0.74}\text{B}_{0.26}$ coating can resist high-temperature to 3000°C . Because ZrO_2 or TiO_2 melting point is below 3000°C . In UHTC system, for $\text{ZrB}_2\text{-SiC}$, can hardly resist to 2800°C in the oxyacetylene flame environment.

As the referee points out, few UHTC materials can resist an oxyacetylene flame environment to 3000°C . So we have to find a new material that can survival such a severe environment for longer times, compared to the conventional materials. Generally, there are two main ways to improve the ablation resistance of materials. One is to find a material and its oxide layer covering substrate having good adhesion to the substrate at high temperature, and therefore the protective oxide layer cannot be easily blown away by the flow gas. The other way is to find a material and its oxide that has a lower oxygen permeability which results in less oxide formed, as well as indicating less loss of oxide would happen under the scouring of hot gas.

In this work, $\text{Zr}_{0.8}\text{Ti}_{0.2}\text{C}_{0.74}\text{B}_{0.26}$ similar to the zirconium carbide would be expected to have a melting point in excess of 3000°C , and its oxide ($\text{Zr}_{0.80}\text{Ti}_{0.20}\text{O}_2$) containing the boron and silicon atoms should have much lower oxygen permeability, compared to the ZrO_2 . For instance, the oxygen diffusion coefficient (ODC) of TiO_2 is $1.12 \times 10^{-13} \text{ m}^2/\text{s}$ at 1800°C , whereas the ODC of ZrO_2 is $1.16 \times 10^{-12} \text{ m}^2/\text{s}$ which is about 10 times higher than that of TiO_2 .

Hence, less $Zr_{0.80}Ti_{0.20}O_2$ covering the $Zr_{0.8}Ti_{0.2}C_{0.74}B_{0.26}$ would form during the same time, indicating less oxide would be blown away by the flow gas, though the melting point of oxide ($Zr_{0.80}Ti_{0.20}O_2$) is lower than 3000°C. That is to say, a low oxygen permeability of $Zr_{0.80}Ti_{0.20}O_2$ leads to less formation and loss of oxide, and further improves the ablation resistance of new material.

Additionally, TiO_2 has a much lower vaporization rate (VR) at high temperature. For instance, the VR of TiO_2 is 0.23 mm/s at 2227°C, whereas the VR of SiO_2 (207 mm/s) is 900 times higher than that of TiO_2 at the same temperature, indicating that ZrO_2 - TiO_2 system from the oxidation of $Zr_{0.8}Ti_{0.2}C_{0.74}B_{0.26}$ would result in much less loss of oxide because of vaporization and good adhesion of oxide layer to substrate shown in Fig.6 (g) even at 3000°C, compared to ZrO_2 - SiO_2 system from the conventional ZrB_2 -SiC and ZrC-SiC.

In conclusion, this new carbide has been refined from the conventional UHTCs through the above strategies. Their ablation results including the mass ablation rate (MAR) and linear ablation rate (LAR) have been improved compared to the other ceramics at 3000°C, though its MAR and LAR increase with the increasing temperature from 2000°C to 3000°C. Consequently, it is believed that the new carbide displaying a better ablation resistance than the conventional ceramics can increase the survival time of components at temperatures up to 3000°C. Therefore, tackle the reviewers concerns, we have modified the text in the “Discussion” section as indicated by the red text shown below and in the revised manuscript.

“For instance, the vaporization rate (VR) of TiO_2 is 0.23 mm/s at 2,227°C, whereas the VR of SiO_2 (207 mm/s) is 900 times higher than that of TiO_2 at the same temperature³⁸. This suggests that oxidation of $Zr_{0.8}Ti_{0.2}C_{0.74}B_{0.26}$ will result in much less loss of oxide because of vaporization and good adhesion to substrate shown in Fig.6g even at 3,000°C, compared to the ZrO_2 - SiO_2 system from the oxidation of ZrB_2 -SiC and ZrC-SiC. Secondly, the good ablation response is due to the low oxygen permeability (OP) of the oxide layer. In this work, the $Zr_{0.80}Ti_{0.20}O_2$ layer on the quaternary carbide has effectively caught the B_2O_3 and SiO_2 which own very low OP ($OP_{B_2O_3}=8.6\times 10^{-12}$ g/cm·s at 1,000°C, $OP_{SiO_2}= 3.2\times 10^{-15}$ g/cm·s at 1,000°C)³⁹ and, to a certain extent, can prevent from the fast diffusing of oxygen atoms into $Zr_{0.8}Ti_{0.2}C_{0.74}B_{0.26}$ at different ablation temperature. Moreover, it is believed that the intrinsic

oxygen diffusion coefficient (ODC) of the $Zr_{0.80}Ti_{0.20}O_2$ layer is lower than that of the pure ZrO_2 due to the lower ODC of TiO_2 ($1.12 \times 10^{-13} \text{ m}^2/\text{s}$, at $1,800 \text{ }^\circ\text{C}$) compared with ZrO_2 ($ODC_{ZrO_2} = 1.16 \times 10^{-12} \text{ m}^2/\text{s}$, at $1,800 \text{ }^\circ\text{C}$)⁴⁰. The lower oxygen permeability of $Zr_{0.80}Ti_{0.20}O_2$ leads to less formation and loss of oxides, and further improves the ablation resistance of carbide, though the melting point of oxide ($Zr_{0.80}Ti_{0.20}O_2$) is lower than $3,000^\circ\text{C}$. Consequently, it is believed that the new carbide displaying a better ablation resistance than the conventional ceramics can increase the survival time of components in extremely oxidizing environments up to 3000°C .”

(3) In the section of ablation mechanism, the authors had better supply cross-section morphology after ablation. It can clearly see the change and variation of oxide layer after ablation.

Following the referees comments, the cross-section morphology after ablation have been investigated by SEM and EPMA. Some interesting observations under the surface have been made and are illustrated in Fig. 6f and g in the revised manuscript. Consequently the Results section is modified as indicated in red text.

“At a higher ablation temperature (2500°C), the $Zr_{0.80}Ti_{0.20}O_2$ crystals connected by the melt become larger and the holes evidently shrink as shown in Fig. 6b. Its cross section shows the porous morphology under the dense surface, as shown in Fig. 6f. Obviously, three different layers can be observed on the cross-section: porous external layer, intermediate layer and dense inner layer, possibly due to the evaporation of oxides and thermal gradient perpendicular to oxide layer. For instance, higher temperature would result in more severe evaporation at the position closed to the external surface. However, the size of pores in layers decreases significantly from the external layer to the inner layer. Especially, the inner layer composed of the grains skeletons and amorphous phases displays a very dense morphology. It is inferred that the dense surface and inner layer acts as a barrier to resist oxidation and results in reduced loss of oxide and the best ablation resistance shown in Fig. 1. At $3,000^\circ\text{C}$, an almost fully dense oxide layer, mainly composed of $Zr_{0.80}Ti_{0.20}O_2$, ZT and SiO_2 , is formed as shown in Fig. 6c, e. Additionally, more amorphous phases occurred according to the variation of the shape and intensity of XRD peaks, with the ablation temperature increasing from $2,000$ to $3,000^\circ\text{C}$, due to

the quenching of more liquid-solid phases of oxide layer at the end of ablation test (some oxides under surface may be solid phases at 3,000 °C within limited ablation time, due to the thermal barrier of Zr-O-Ti ceramic system³⁷). Meanwhile, the sealing of the oxides can be attributed to these melts having a relative lower viscosity. However, such melts seal the defects (holes and cracks arising from the ablation) and protects the carbon matrix well, causing greater loss of the oxides by the scouring of hot high-speed gas as well, which is confirmed by the relatively low weight gains shown in Fig. 1 and the corrosion pores beneath the dense surface shown by the cross-section morphology of the sample after 3,000 °C ablation in Fig. 6g. However, a relative dense inner layer located at the interface between oxide layer and carbide acted a barrier to the diffusion of oxygen, though some pores occurred in a thicker oxide layer formed with the temperature increasing from 2,500 to 3,000 °C. Additionally, micro-cracks occurred on the cross-sections due to the thermal shock during the ablation test. The relatively integrated interfaces between oxide layers and carbides experiencing 2,500 and 3,000 °C ablation tests, showing a compact morphology without separation, indicate a good adhesion of oxide layers to substrates. Notably, it is believed that the dense $Zr_{0.80}Ti_{0.20}O_2$ layer has effectively retained the boron and silicon and extended their consumption time, according to their residual contents shown in Fig. 6d, f. ”

Figure 6 Microstructure and phases of ablated surface and cross section. f, Cross-section morphology of central ablated point at 2,500°C (back scattered electron images), and the associated distribution of elements. g, Cross-section morphology of central ablated point at 3,000°C.

(4) With the increase in ablation temperature, more amorphous phases occurred

according to the variation of the shape and intensity of XRD peaks and SEM. Why these phenomenon (amorphous phases) happen?

Correction. It is because more liquid phases will form on the ablated surface with the increase in the ablation temperature. Especially, 3,000°C is beyond the melting point of the oxides, indicating more liquid phases occurred. However, some oxides under the surface may still be solid phases due to the thermal barrier of ZrO₂ within the limited ablation time. Therefore, the oxide layer should be a liquid-solid phase during the test. When the test finished and the ablation temperature decreased to the room temperature shortly, little crystallization of oxides on the surface of layer occurred, indicating a quenching process happened which caused the formation of amorphous phases. Consequently, we have added the following contents in “protective mechanisms” of the revised manuscript.

“Additionally, the XRD results suggest the presence of more amorphous phases, with the ablation temperature increasing from 2,000 to 3,000°C, due to the quenching of more liquid-solid phases of oxide layer at the end of the ablation test (some oxides under the surface may remain solid at 3,000 °C within the limited ablation time, due to the thermal barrier provided by the Zr-O-Ti ceramic system³⁷).”

Reviewer #2 (Remarks to the Author):

I believe this paper will spark interest in investigating other quaternary compounds for hypersonic applications as it represents a “new” promising material class.

I believe the work of the authors is well described in this overall excellent paper. Not merely the presentation of some interesting results, the authors perform a relatively in depth analysis of the new compound. For example, see Figure 6c and d, where atomic placement is shown.

Many thanks for your supporting comments, which encourages us greatly.

Some comments:

1. Page 3, line 75 – there is no matching parenthesis.

We have corrected it in the revised manuscript.

2. Page 4 – what is the heat flux of the ablation flame? This is a standard way to compare data. Also, what opening was used on the gas tip? 3 mm dia? 5 mm dia? Why did the tests last “approximately” 60 secs and 120 secs as opposed to exactly 60 secs or 120 secs? What was the error in time measurement?

Sorry for omitting this information which we have corrected. Actually, the time is calculated automatically by computer. It is highly accurate (0.001s) and so we have deleted the “approximately”. Additionally, we have updated the ablation information in the Methods section of the revised manuscript, as shown in the following content.

“The heat fluxes measured by a water-cooled heat flux sensor at 3000, 2500 and 2000 °C is 5.62, 3.86, 2.57 MW/m², respectively. The inner diameter of the oxyacetylene gun tip was 2 mm.”

Reviewers' Comments:

Reviewer #1 (Remarks to the Author):

This revised manuscript has been corrected according to the reviewer's suggestion, and can be accepted to publish.

Reviewer #2 (Remarks to the Author):

The authors have made substantial improvements to the manuscript. I recommend for publication.

Response to Reviewers:

Reviewers' Comments:

Reviewer #1 (Remarks to the Author):

This revised manuscript has been corrected according to the reviewer's suggestion, and can be accepted to publish.

√ Many thanks.

Reviewer #2 (Remarks to the Author):

The authors have made substantial improvements to the manuscript. I recommend for publication.

√ Many thanks.